# Barriers to accessing high-quality cancer medicines in Cameroon. A qualitative study of the views and practices of regulators and frontline healthcare providers

Yauba Saidu[1,2], Masong Christine Makia[3]*, Andreas Frambo[1], Armstrong Njuh Nang[1], Atalay Mulu[4], Ayenew Ashenef[4], Muluneh Benyam[5], Ibrahim Chikowe[6], Sonak D. Pastakia[7], Colleen R. Higgins[5], Ndom Paul[8], Marya Lieberman[9]

**1** Clinton Health Access Initiative, Yaoundé, Cameroon, **2** Institute for Global Health, University of Siena, Siena, Italy, **3** Department of Social Sciences and Management, Catholic University of Central Africa, Yaoundé, Cameroon, **4** Department of Pharmaceutical Chemistry and Pharmacognosy, School of Pharmacy, College of Health Sciences, Addis Ababa University, Addis Ababa, Ethiopia, **5** Department of Pharmacy, University of North Carolina, Chapel Hill, United States of America, **6** Department of Pharmacy, Kamuzu University of Health Sciences, Blantyre, Malawi, **7** Department of Pharmacy Practice, Purdue University College of Pharmacy, Indianapolis, United States of America, **8** Faculty of Medicine and Biomedical Sciences, University of Yaoundé 1, Yaoundé, Cameroon, **9** Department of Chemistry and Biochemistry, University of Notre Dame, Notre Dame, United States of America

* masongbye@yahoo.com

## Abstract

Cancer has become a major public health problem in Cameroon. In 2020, 20,000 new cases and 14,000 deaths were registered. Despite its high burden, access to quality medications for this condition continues to be a persistent challenge in the country. This study assessed the perceptions and practices of key stakeholders on barriers to accessing quality anticancer medicines in Cameroon. The overall goal was to generate data on key challenges related to the quality of anticancer medications and their impact on cancer care and treatment in Cameroon. This is to inform policy and interventions aimed at addressing the problem. In a cross-sectional qualitative study, a document review of key reports and documents, and 57 stakeholders were purposively identified and interviewed. The choice of these respondents was informed by their engagements in one of the areas of the chemotherapy supply chain in Cameroon, including regulation, procurement, quality control, and use. Data from document notes and interviews were analyzed using a thematic approach. Several factors were identified as affecting the quality of chemotherapy medications, including weak regulatory systems, inadequate funding for regulatory activities, and a lack of core competencies among staff responsible for regulatory functions. Other factors included the high cost of anticancer medications, which influenced accessibility to quality products, as well as the quality of care offered to patients who could not afford high-priced products. These weaknesses, coupled with a lack of point-of-care tools, appear to encourage the proliferation of substandard medicines in our setting and the

**Data availability statement:** All data underlying study findings is reported within the paper.

**Funding:** Research reported in this publication was supported by the National Cancer Institute of the National Institutes of Health (Award Number U01CA269195 to ML). The funders had no role in the study design, data collection and analysis, decision to publish, or preparation of the manuscript. The content is solely the responsibility of the authors and does not necessarily represent the official views of the National Institutes of Health.

**Competing interests:** The authors have declared that no competing interests exist.

use of substandard therapies or protocols to treat patients. Our study sheds light on the multifaceted problems that plague Cameroon's chemotherapy supply chain and how these impact access to optimal care and treatment for cancer patients. Urgent actions are needed to enhance the regulatory landscape and improve the affordability of quality anticancer medications in Cameroon.

## Introduction

Noncommunicable Diseases and Injuries (NCDIs) have emerged as a major public health threat globally, leading to about 41 million deaths [1]. The majority of these deaths occur in low and middle-income countries (LMICs), where these conditions disproportionately claim the lives mostly of people aged between 30 and 69 years old [1], causing both an economic and social loss for these countries [2]. Cancer alone accounted for over 10 million deaths in 2022, with over 70% of these occurring in LMICs [1]. Despite this high mortality, the survival rate of patients with cancer in LMICs is 30%-50% lower than that of patients in high-income countries [3,4]. This high mortality rate in LMICs may reflect multiple factors [5], including advanced-stage at diagnosis due to late presentation [6]; inadequate infrastructure and health work force to handle the condition [7]; cultural dynamics and perceptions affecting treatment and diagnosis [8–10]; and inadequate access to quality-assured medicines [11–13].

Access to anticancer medications is a key challenge in LMICs, with reports showing that only approximately 30% of patients diagnosed with cancers have access to quality anticancer medication [10]. Quality medication has been described as encompassing the comprehensive attributes of a pharmaceutical product, ensuring its safety, effectiveness, and being free from contamination or defects; involving evaluating the product's purity, impurities, component content, and physical/chemical properties to guarantee it meets established standards and provides its expected therapeutic effect [14]. In Cameroon, the quality of pharmaceutical products is influenced by several factors. These include regulatory challenges, such as weak enforcement and oversight, which hinder effective drug quality control; fiscal constraints that impact the national health budget; and a high centralization of cancer services, as well as workforce shortages, which negatively impact optimal cancer service delivery [7,15]. In addition to these, inefficient drug procurement and distribution lead to frequent stockouts of essential cancer medicines in treatment centers [7]. The scarcity of quality products not only interferes with the timely treatment of patients following diagnosis [16] but also opens an avenue for the conscious or unconscious use of substandard or falsified treatments [17]. This invariably results in several untoward health, societal, and economic consequences [18], especially in children [19,20].

Cancer is a major public health concern in Cameroon. In 2020, over 20,000 new cases and 14,000 cancer-related deaths were registered in the country, with this reported as an underestimation [21]. With this growing burden of cancer morbidity and mortality, Cameroon faces an uphill task in expanding cancer care and treatment

to the patients and families who desperately need it. The country currently has only seven (mostly low-resourced) cancer care and treatment centers, run by the state (state-owned hospitals) and by private bodies (faith-based and non-faith-based cancer specialist centers), which offer chemotherapy services and palliative care to a population of nearly 30 million people [7]. Most state treatment/care centers have a co-pay plan for treatment, which is useful when the patient has an insurance policy (usually a minority) or, in the absence of an insurance policy, is paid fully out of pocket by the patient and their family. [7] In state-owned hospitals/care centers, the State offers subsidies for certain anticancer medications sold within their pharmacies, reducing their cost. However, when these drugs are not in stock, patients must visit private pharmacies themselves and purchase the drugs at a higher price [7]. The sources of anticancer medications were pharmaceutical companies, which held the largest share (approximately 43%), followed by the National Drug Procurement Agency (29%) and private pharmacies (14%). State-owned hospitals typically had the most limited stock [7].

The local cancer supply chain (a term which we use here interchangeably with cancer supply system, to include the entire infrastructure and processes involved in getting anticancer medications to patients, including aspects like procurement, storage, and dispensing; and the roles of regulation and control which these stakeholders have) is usually unpredictable, with frequent stockouts of these lifesaving anticancer medications. This leads to questions about the quality of the medicines that patients use, as during stockouts, patients and clinicians are charged with finding alternative means to procure medicines, raising concerns over the quality of these medications, and whether patients might not receive the intended dose or active ingredient for these medications with a narrow therapeutic index. Despite these, there is limited literature on the quality of anticancer medicines circulating in the Cameroonian market. Tchounga et al (2021) have reported a high prevalence of substandard or falsified (SF) treatments for many molecules, anti-inflammatory and antibiotic medicines in Cameroon [22], but though their study did not explore anticancer medications, it sends a strong signal about the possibility of SF treatments in Douala and Yaoundé, the sites of the leading cancer treatment centers in Cameroon. The findings in this study provide a rationale for investigating the presence and better understanding the different perspectives on SF cancer medications within the Cameroonian context. Leveraging such findings could be useful in developing user-informed interventions in addressing a holistic approach to managing cancer, by targeting drug quality, which is becoming a public health challenge. This qualitative study was therefore designed to assess existing perceptions and practices related to anticancer drug quality among frontline healthcare workers and regulators in Cameroon. We provide an overarching summary and highlight opportunities for strengthening quality control for these drugs here.

## Materials and methods

### Study design and setting

This was a cross-sectional, qualitative study conducted in two cities in Cameroon (Yaoundé and Douala) between October 9th and 25th, 2023. Semi-structured interviews were conducted with 57 key informants, all of whom were stakeholders involved in national cancer treatment efforts. We also reviewed key administrative reports to provide key background information, triangulate data, and substantiate claims made by interviewees. Saturation was reached once the information received was similar, amongst participant groups. In the case of regulatory bodies, their objectives and roles were similar, so we focused on speaking only with regulators directly involved in anticancer post-marketing drug testing (this limited the number of respondents interviewed among regulators).

### Sampling and participant selection

Data collection was based on purposive sampling, where key informants were selected based on characteristics relevant to the study, such as their place of work and skill training, those considered the most informative for the study, and those who were also available to respond. When a possible informant was unavailable for an interview, snowball sampling was employed, and another informant with similar characteristics was proposed by this or another key informant. The

respondents targeted were made up of three main groups: (1) regulation and quality control officials; (2) procurement personnel; and (3) health-care personnel. The regulation group comprised the Department of Pharmacy, Laboratory, and Medicines, the National Cancer Control Program, and the National Drug Quality Control and Evaluation Laboratory (all these making up the national regulatory body within the Cameroon Ministry of Public Health). These are the key technical regulators responsible for quality testing and control (post-marketing). They all work interchangeably in the case of anticancer medications. The procurement group consisted of actors within the chemotherapy supply system, made up of public and private pharmacies, distributors, and the National Procurement Center for Essential Medicines and Medical Consumables (CENAME). The group with healthcare personnel comprised oncologists and oncology nurses in public hospitals and private cancer treatment centers.

After purposively identifying participants, we adapted and utilized interview guides originally developed to assess the situation of anticancer medicine quality from multiple perspectives in the fields of cancer treatment, medicines supply, and medicines regulation in LMICs, to conduct in-depth interviews with key actors. These interview guides (S1 Text, S2 Text) were designed to elicit the practices, knowledge, and behaviors of frontline cancer control health workers regarding the quality of anticancer medications and the accessibility of these medications to cancer patients. A limited number of questions were directed at respondents' perceptions, although these perceptions were further teased out of responses to factual questions. A pilot study was conducted with the interview guides in Yaoundé on one (1) respondent of each participant group (except for regulators) to test usability in terms of understanding, the length of time needed with participants, and any adjustments integrated (the pilot interviews were not considered within this study). Upon obtaining their informed consent, 57 key informants were interviewed. The interviews lasted between 35 and 60 minutes, and two interviewers conducted each. While one person interviewed, the other took notes in addition to the audio recordings. All interviews were initially recorded and transcribed in French, and subsequently translated into English.

Documents reviewed were reports, strategic documents, and blogs purposively identified and collected from administrative websites related to the management of medicines in Cameroon. These were selected from the websites or webpages of the Directorate of Pharmacy, Medicines, and Laboratories [23] and the Cameroon Ministry of Public Health website [24]. In addition, unpublished national cancer control documents (national strategies and descriptions, reports) [24,25] were considered. The review was conducted alongside interviews, adding depth to most of the participant responses and assisting in verifying the validity of statements as part of data triangulation.

### Analysis

All recordings were first transcribed verbatim, and the transcripts were then translated from French (the original language of data collection) into English and subsequently anonymized. Document notes and transcribed data (transcripts) were then managed manually. Subsequently, the qualitative data analysis tool NVivo 14 was used to organize the data and codes developed from these. Based on a thematic analysis [26], an initial familiarization with all data was conducted, followed by the generation of an initial line of iterative coding, which was then developed into main themes. Interview guides and study objectives were reviewed to ensure that no points were left out during code development and later theme development. The thematic frame for coding was based on a structural-functionalist perspective, where, within this framework, the role of each actor or participant group was considered a structure within the healthcare landscape for cancer control. These different structures were then probed to understand how they each organized to control the quality of chemotherapy drugs before and after sales to patients, as independent actors within the health system. Additionally, this perspective afforded the possibility of considering the different structures within the drug distribution system, with an emphasis on how they each function to enhance or hinder access to quality chemotherapy medicines. Each coding process was verified across the entire study team before being registered. Main themes were generated and interpreted to provide the key results for this study.

Quantitative data, such as participant demographics, were entered into Excel, and the resulting tabulations and figures were generated.

**Ethical clearance and informed consent**

Approval for this study was obtained from Cameroon's National Ethics Committee on Human Health Research (Ref N° 2023/01/1517/ CE/CNERSH/SP). This approval was supplemented by administrative authorizations from Cameroon's Ministry of Health as well as the Regional Delegations for public health in the Littoral and Centre Regions. Written informed consent was further obtained from all participants before each interview. All participants were above 18 years of age.

## 3. Results

**Baseline characteristics**

A total of 57 respondents were interviewed. These included: one (1) drug regulator (from the national technical drug regulating body), five (5) drug distributors (private wholesalers involved in supplying anticancer medications), twelve (12) oncology nurses, fourteen (14) medical oncologists, and twenty-five (25) public (practicing within public hospitals) and private retail pharmacists handling oncology/anticancer medications. Most of the study respondents acknowledged there was a big challenge with the quality of anticancer medicines in the Cameroonian setting. Similarly, respondents unanimously identified affordability and availability of these anticancer medicines as a veritable challenge for patients and their families. Oncology nurses, oncologists, and drug distributors all acknowledged the presence of SF medications as a reality and a problem for anticancer treatment, whereas, for the most part, pharmacists believed that most of the challenges were related to packaging mistakes rather than the quality of the medicines. Over 70% (19/25) of pharmacists did not consider packaging irregularities as a quality concern.

**Factors affecting the quality of chemotherapy medicines in Cameroon**

Several factors were identified as interfering with the quality of chemotherapy drugs, hindering their objective of providing safe and effective cancer management in Cameroon. These included existing structural and regulatory barriers in ensuring the quality of these drugs before they are put on the market, as well as barriers in accessing these drugs by patients and their families, driven by issues with their availability and prohibitive costs. Also, the impact of poor-quality drugs on treatment regimens and outcomes was raised as a key challenge in the health service provision for cancer.

   1. **Structural elements for quality control of anticancer drugs in Cameroon.** We identified four governance structures involved in the anticancer supply chain in Cameroon, outside of pharmacies [23] (Fig 1). These structures work together and, at some point, share the same general objective (regulation) through complementary activities. Among these, the National Department of Pharmacy, Drugs, and Laboratory is the first, serving as the main drug regulatory body responsible for licensing new medicines and issuing marketing authorizations. This structure also serves as the main administrative body, supervising the implementation of drug regulations in Cameroon, as well as the release of lots. It works in close collaboration with the second structure, the National Drug Quality Control and Expertise Laboratory (LANACOME). The LANACOME supports the National Department of Pharmacy, Drugs, and Laboratory with post-marketing technical analysis of drug samples to ascertain their qualities before making a recommendation to approve or reject a marketing authorization application. Once approved, the third structure, the National Cancer Control Program (NCCP), accompanies this by overseeing the development and implementation of cancer strategies and policies in Cameroon, supporting public health facilities in procuring these medications. The NCCP functions through quantification and forecasting, as well as "lobbying the state to allocate and mobilize resources for this purpose" (Regulator #1). The NCCP then forwards the quantities identified for procurement to the National Agency for Essential Medicines and Consumables (CENAME), which mobilizes the necessary resources to procure the drugs from manufacturers, who are primarily based in India. Once these have been procured, the NCCP and CENAME coordinate the distribution of these medications to the seven (7) public cancer treatment centers in Cameroon. Although this is supposed to be a standard

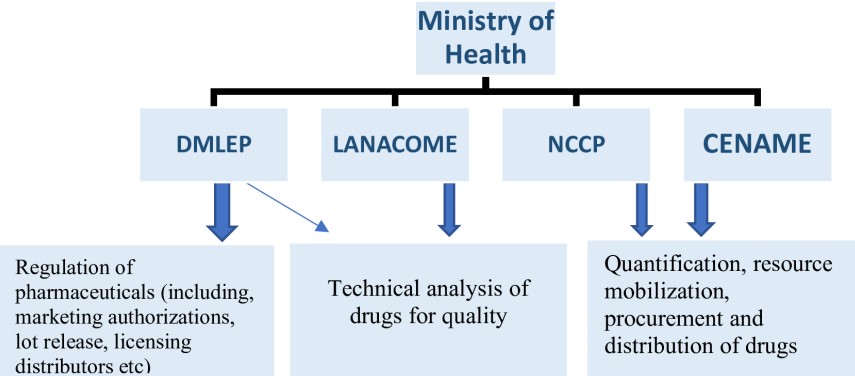

**Fig 1. Core structures for governance in the supply chain of anticancer drugs (National Department of Pharmacy, Drugs and Laboratory (DMLEP), National Drug Quality Control and Expertise Laboratory (LANACOME), National Cancer Control program (NCCP), National Agency for essential medicines and consumables (CENAME).**

annual process, "both the NCCP and CENAME have not been able to formally and sufficiently acquire and distribute anticancer medicines for the past three years or so, due to limited funds" (Regulator #1). In these previous years, "the quantities acquired were quite insufficient, primarily due to poor forecasting as well as inadequate mobilization of the necessary financial resources from the state" (Regulator #1).

Although these bodies were found to be very engaged in their administrative procedures, with constant checks and site visits, the following gaps were identified by our key informants (regulators and distributors): insufficient funds and autonomy of CENAME and NCCP to procure and supply enough drugs to meet demand, and ensure price controls across all distributors/pharmacies; limited technical capacity and lack of (practical) financial autonomy of LANACOME hinders its ability to adequately assess the drug quality of circulating cancer medications; and cumbersome procedures (e.g., lengthy durations) for cancer drug licensing and procurement often lead to issues with the adequacy, availability, and affordability of anticancer medicines.

The technical and financial capacity of LANACOME to execute its functions was raised by respondents as a key obstacle to assuring the quality of anticancer medicines in Cameroon. This obstacle was aligned to those described by LANACOME in its 2023 annual report, [MINSANTE,2023] where it described several bottlenecks in its functioning such as "insufficient materials for product evaluation"; "insufficient reagents and standard reference preparations"; "inadequate test samples received by the laboratory"; "lack of key equipment for analysis"; "lack of key normative documents", "unmet capacity building needs for technicians". This, for example, hinders the capacity of technicians to perform their technical duties, as noted by a regulator in Yaoundé:

*"Certificate of analysis, …, it's a voluminous file, so sometimes you have to do ten[certificates], [but] the money we [receive] so that LANACOM does its part…[is insufficient]. So, after a year or a year and a half, you have to repeat the survey tests and reassess to see what that reveals. This exercise is rarely done, [because] the resources made available to LANACOME... are not a lot" Regulator #1, Yaoundé*

Additionally, delays in the public drug procurement process were identified as a challenge. When the National procurement agency faces treasury changes, this limits its ability to pay manufacturers before importing medications. Thus, the government issues tenders, which private pharmacies can compete for. In the instance that they obtain a "good" in the evaluation process [notably the technical (70%) and financial position (30%)], they can be/or are awarded the contract to import the medications and supply them to the government via the National procurement agency. This latter then

distributes the medicines to the cancer treatment centers. Typically, a tender to procure medicines could take "several weeks or months to be approved" (Distributor #1). And once approved, the file would need to go through several ministries before a distributor is selected. This lengthy process contributes to stockouts of some drugs at various service delivery points, which in turn causes a hike in prices in private pharmacies with these drugs, and at the same time opens a risk for cheaper low-quality drugs to filter in.

**2. Challenges related to irregular or suspicious chemotherapy medicines.** Suspicions around the quality of these cancer medications existed amongst health workers in the study. Key informants reported that drugs dispensed or administered to patients were suspected to be substandard or falsified. Other than visually assessing the packaging or expiry dates, these health workers reported not being equipped with tools or methods to verify the quality of the medication before administering it to patients. They pointed out that they can only suspect the poor quality of the drug after observing a deterioration in the patient's state of health after treatment initiation:

*"…their [cancer patient taking chemotherapy] health deteriorates instead of improving, then we start seeing[considering] it could be the medicine, it is not good quality..." Oncology Nurse #1, Yaoundé*

*"…I have had several cases with patients here, I give certain medications here, people develop certain toxicities which are not expected or normal to the drugs, I was traumatized here during a period when I had three or four deaths that I could not explain" Oncologist #1, Yaoundé*

*"The people had chemo, then serious hematological toxicities to the point where I asked the Matron to take a plastic bag and take all the boxes of medicines from all the patients and to look where the people bought the drugs and what boxes they were…these are things that we have to be quite vigilant about" Oncologist #2, Yaoundé.*

These quality concerns have prompted several healthcare providers to adopt proactive approaches, systematically examining the packaging of medications brought to the bedside before administering them to patients. For many healthcare workers, inspecting the packaging is the only tool they have, which can provide them with an indication of the quality of the chemotherapy drug, as posited below:

*"I already recognize the boxes of medicine that will bother me… today the patient has a bottle where it is powder and not diluted, tomorrow he has a bottle where it is already diluted. After tomorrow, the medication is more of an aqueous substance, where you have to apply pressure before [opening it], it doesn't even make it easier for people[young doctors or nurses] who are learning…So for me, we need to reframe that. Let us know that this[packaging] is what we have, this is the medicine, which has the right packaging, that we are using" Oncologist #3, Douala*

*"…there are certain aspects of packaging [differing from the norm] causing problems… IDI, Oncologist, Douala*

*"it is the expiry date we are mostly[supposed] to look at, but with our experiences, now we start looking at the color too and other things, because of what we have seen before…it is things you do not expect to see. Different colors for the same medicine, and things like what…" Oncology nurse #4, Douala*

**3. Challenges linked to the accessibility of quality chemotherapy medicines.** Inaccessibility to quality anticancer drugs was a strong point raised by all respondent groups. According to respondents, the high prices of these medicines render them unaffordable to many patients and families. This was also a similar situation in public pharmacies (those located within public hospitals), which offered these products at subsidized rates. In many instances, these subsidies at public pharmacies often resulted in a demand that outstripped supply. The demand, coupled with lengthy procurement bottlenecks as well as the inability of the State to mobilize resources promptly, often created prolonged stockouts of these

products in public hospital pharmacies, which in turn left the patient and healthcare providers in precarious situations as summarized below:

> "A patient arrives today for these products, tomorrow he comes back, and he doesn't get any. That's it. And you know what it does morally, psychologically to a patient, who has started his treatment, [or] who has wanted to start his treatment and he is no longer able to take his treatment, not because he has no money, because the product is no longer available, we can't find it anywhere" Oncology Nurse #9, Yaoundé

The high costs in our setting not only limit access to these lifesaving products but also create a gap that enables the uncontrolled circulation of SF products, which patients could afford at lower prices.

> "…you have to know, many patients are looking for health, sometimes also at lower prices…many people often tend to look for the cheapest. That's the problem. And many drugs that do not respect the distribution rules are generally cheaper and in the corridors [grey markets], sometimes, not very clear, very often it is by word of mouth. Someone will tell you Oh I have someone who can get you this medicine." Oncologist #12, Douala

Additionally, the unavailability of these products sometimes resulted in a "survivalist environment" where patients went to great lengths to procure these medicines, even when they were unregistered, or "imported" them from dubious sources, as posited by the respondent below.

> "Unfortunately, the products aren't available. So…they look [for them] elsewhere, and you cannot stop them because they need to be treated…if we cannot bring it [the drugs] to them, they have to look for [the products] themselves…" Distributor #2, Douala

This inaccessibility to chemotherapy medications was a source of distress to many health personnel in the surveyed facilities, at times causing them to take actions intending to alleviate patient suffering. As such, respondents acknowledged occasionally administering medicines of dubious quality in an attempt to "help" patients. Some personnel even went further to save up incompletely used (or already opened) medicines from a previous patient to "assist other patients to complete their doses if needed" (Oncologist #5). This practice of preserving "leftover" medicines to assist other patients who can access the products raises several questions, which further amplify the uncertainty of the quality of medicines being administered to patients in Cameroon.

Furthermore, although healthcare providers reported being impacted by the low accessibility of quality cancer medications, the challenge was mostly felt by the patients: they had to bear the psychological and financial stress in addition to their ill health. Even when patients could afford it, they could not find the medications in the local market. This accessibility challenge was recognized as one of the principal causes of treatment dropouts, increased cancer-related deaths amongst patients, and health worker demotivation. One respondent felt that the high cost of cancer treatment kills more patients in Cameroon than the underlying cancer.

> "…they say in the quarter [within communities] cancer kills…this needs to be changed…it is the cost of cancer treatment that kills the patient in Cameroon. Not the disease on its own. When they get diagnosed earlier and take the treatment prescribed for them at the right intervals, the chance of survival is high…you can see in the developed countries. Cancer in a state of poverty, as is the case in our contexts, is almost equivalent to a death sentence. No money, no drugs, societal stigma…" Program personnel (Procurement)#1, Yaoundé

**4. Unstandardized treatment protocols affect overall cancer care and treatment.** Lastly, the availability of anticancer medications was found to significantly influence the treatment regimens and protocols adopted by healthcare

practitioners in the region. In many instances, treatment plans were determined based on the accessibility and affordability of specific drugs, resulting in significant variations in treatment approaches for similar types of cancers. This situation fostered inequity and a lack of standardized practices in cancer treatment within the cancer treatment sites. The unstandardized pricing and increased stockouts resulting from these could also impact quality.

> *"Financially, AC [Doxorubicin and Cyclophosphamide] regimen for breast cancer is best[for the patient]…the treatment interval is 21 days. But with Taxol, it's every week. So, at 21 days, they have time to prepare for the money… we use the AC mostly because of this." Oncologist #7, Yaoundé*

> *"Yes… they cost a little more. Even if they are present, the patients...certain patients, especially those who have already received 2,3,4 chemotherapies, are already financially exhausted. It's no longer accessible enough; there are always alternatives because there are slightly more expensive protocols. We try to find alternatives[they can afford] for them" Oncologist #8, Douala*

Key informants noted that there was a risk of high circulation of substandard versions of anticancer medications due to high demand, raising the need for heightened vigilance for such medicines. In this study, four molecules, notably Cisplatin, Oxaliplatin, Doxorubicin, and Cyclophosphamide, were identified as the "most prescribed" and the "most available". Despite their high availability, oncologists and oncology nurses who manipulate these molecules on a daily basis signaled "variances in their packaging" and cast doubt on the "quality based on patients' reactions to these products" (Oncologist #5 Yaoundé), especially for Doxorubicin and Cisplatin. Respondents believed there was a high probability that these products would be substandard or falsified due to their high demand. This claim was based on the fact that respondents have noted "suspicious" changes in the packaging of these products, as well as an unusual "coloring" of the products at the point of administration.

> *"Sometimes it's also in the composition itself. Sometimes there is a color problem. For example, we know that doxorubicin is a drug that turns red when diluted. Sometimes, you can have doxorubicin that appears white when you dilute it. That raises questions about the quality of the product" Oncology Nurse #11, Douala*

At all levels, the difficulties expressed regarding the lack of financial support for cancer care and treatment efforts, as well as the flawed organization of anticancer drug procurement schemes for both private and public pharmacies in Cameroon, showed to encourage the presence and use of SF cancer medications. Efforts addressing these challenges would go a long way to improve not only on the availability and affordability of anticancer medications, but their quality as well.

## Discussion

This study sheds light on the complexities surrounding access to quality anticancer medications. The lack of a predictable supply chain for anticancer medications and the apparent absence of subsidy schemes for these products in many settings raise fundamental questions about the quality of oncology products circulating in countries with weak regulatory systems [16,27]. This challenge is becoming an important thematic area for global public health as several studies have reported a high prevalence of illicit oncology drugs within legitimate supply chains in many countries across the globe [28,29]. This high prevalence is in part driven by the high prices of these products, which make them attractive targets for falsification. In 2016, these products were ranked as the fifth most commonly falsified molecules in the world [30].

The findings from our study, the first from Cameroon, shed light on the complexities surrounding the quality, availability, and affordability of anticancer medicines in this part of the world. Here, we identified three key factors that contribute, either individually or synergistically, to stockouts as well as the proliferation of SF anticancer medicines in Cameroon. The first critical shortcoming was the limited capacity of the national regulatory authority, which hinders its ability to properly

execute its function of safeguarding public health by scrutinizing the quality, safety, and efficacy of medicines before authorizing their use in the market. This limitation, which also extends to weaknesses in monitoring capabilities, has been shown to stem from several gaps, including inadequate funding of the regulatory system, a lack of quality systems, equipment, reagents, and trained human resources to conduct regulatory functions. These findings align with those reported by Baiye et al. (2022) [15], Ndomondo-Sigonda (2017) [31], and the WHO [32]. Addressing these gaps will therefore require the design and implementation of targeted interventions [33], which should range from enacting new policies and reforms to equipping the regulatory body with sufficient resources, including financial, human, and infrastructural. Such investments, if well implemented, would make the regulatory body "fit-for-purpose", enabling it to provide stringent quality reviews and controls as well as enhanced pharmacovigilance capabilities to detect and manage the circulation of SF medications in the country.

The second critical challenge identified in this study revealed that clinicians working in oncology within the Cameroonian setting frequently lacked the means to verify the quality of suspicious products before administering them to patients. Consequently, many of them rely solely on visual inspections, which primarily involve checking expiry dates and product colors or packaging, for clinical decision-making. This limitation increases the probability of administering SF chemotherapies to patients and invariably contributes to the increased mortality amongst cancer patients in our setting. Addressing this gap would require several interventions, beginning with building the capacities of healthcare workers on the necessity for, and methods of, quality control beyond "packaging assessments" and "inquiries". Clear clinical Standard Procedures are needed in all treatment centers, which should outline a step-by-step procedure for each critical function or task. This could include inventory management, drug conservation, visual inspections, as well as documenting basic data about the drug, such as sources of purchase, manufacturer, lot numbers, findings from "packaging assessments" and visual inspections, amongst others [33]. This system could create a platform that would easily enable follow-up, which could eventually feed directly into existing alert systems, such as the online National Department of Pharmacy, Drugs, and Laboratory alert platform [23].

In addition to establishing clear and standardized systems within these contexts, the use of point-of-care tools could bring significant relief to oncologists and patients in resource-limited settings. These point-of-care tools are being increasingly used in several high-income settings, including the European Union [33,34]. Many of these tools, such as the Minilab system [35], are expensive (costing approximately $4,000) and mostly impractical for use in Cameroon or other LMICs, necessitating the development of methods for chemo-drug safety and personal protective equipment to protect personnel while handling anticancer medication. This limitation underscores the pressing need for practical and cost-effective tools that clinicians can readily use to assess the quality of anticancer medicines at the bedside. A tool like the "chemoPAD" [36], which is a paper analytical device for screening chemotherapy drug quality, can easily fill this unmet need. Its potential in filling this gap was recently reported from Ethiopia [13], where a proof of concept of its value for the rapid detection of SF chemotherapies at the bedside was established [13]. Expanded evaluations of the chemoPAD's value are currently ongoing in four resource-limited settings in SSA, including Cameroon, Ethiopia, Kenya, and Malawi. While this tool may bring hope to oncologists/nurses and patients, there is still an urgent need to raise awareness about SF medicines among the various cadres of healthcare workers involved in the chemotherapy supply chain in Cameroon, including pharmacists. This is due to the relatively low level of awareness around SF medications amongst actors in the oncology drug supply chain; over 70% of those in private pharmacies interviewed in this study failed to acknowledge the pervasive problem of SF anticancer medications in Cameroon. This lack of acknowledgement contrasts with the experience of oncologists and oncology nurses who have firsthand experience with SF anticancer medicines.

The third challenge we unveiled in our study was linked to the accessibility of lifesaving anticancer medications. The high cost of these medications [29], coupled with the absence of a subsidy scheme, places a significant financial burden on patients and healthcare facilities. The high cost not only impedes patients from accessing these products but also prompts them to seek cheaper alternatives, which are likely to come from questionable sources, as seen in other settings

[29,37]. Like in other settings, these high costs were identified as a major push for patients and their families to discontinue treatment [10], especially in Cameroon, where out-of-pocket costs for cancer treatment were a huge burden on patients and their families [7]. This, in turn, increases the mortality rate from cancers.

Furthermore, we observed a complex interplay between high costs and the quality of care provided to patients by clinicians. Most clinicians reported adjusting treatment protocols based on the availability of specific oncology products, which at times encourages or unknowingly involves SF medicines. This tailoring of treatment protocols creates significant inequities in cancer care within the same facilities and patients with the same cancers, which may invariably impact clinical responses, including therapeutic benefits, occurrences of adverse events, and mortality.

Optimal accessibility to quality anticancer medications has been a longstanding problem in LMICs[38]. This challenge has been described as "market failure" because the economic incentives are not high enough in these markets for reputable pharmaceutical companies to prioritize sales of these high-cost products [11,16]. Despite this limitation, a positive development for cancer patients has gradually emerged in recent years in Cameroon and elsewhere, for example, with efforts to improve access to cancer drugs through access deals negotiated by the Clinton Health Access Initiative and the American Cancer Society, alongside major pharmaceutical companies, including Novartis, Pfizer, Biocon and Roche amongst others [39,40]. Although this has been slow to take off, improved availability and affordability of quality cancer products, as well as patient outcomes, have been reported in Nigeria, Ethiopia, and Uganda [38]. To intensify and expand these efforts across Cameroon and other settings, a strong push would be needed from multiple stakeholders, especially governments and civil society. An engagement of the civil society would also be important to address negative social perceptions of cancer, which has also been shown to interfere with treatment [10]. Their advocacy would also be needed to highlight the need for the availability of cancer care and treatment facilities at all levels, as well as the inclusion of cancer care and treatment in the universal health package within Cameroon, specifically. This advocacy can be a key step in championing the cause for cancer in Cameroon and several other SSA countries, leveraging events like "Pink October" (for cervical cancer), which is heavily promoted by civil society organizations, non-profits, and for-profit organizations, as well as governments. Such advocacy, if well-tailored, can spur positive developments for other cancer types, including childhood cancer, which still faces outstanding challenges in LMICs[19]and in Cameroon [41], as well as raising awareness about the importance of drug quality.

Despite the valuable insights from this study, certain limitations must be acknowledged. The first relates to the limited number of regulars questioned and the lack of cancer patients among respondents. Patients were not targeted as respondents in this study because we aimed to consider the supply of anticancer medications and the quality control carried out by wholesale distributors, private and public pharmacies, as well as regulators. Cancer patients' perspectives would have provided firsthand information on access to, and sources for, medications in and out of treatment centers. Otherwise, oncology nurses who are frontline service providers and have firsthand knowledge of the practices and lived experiences of people living with cancer were a key source of related information within this study. Secondly, our research targeted drug dispensers within the formal supply chain system, which led to our missing critical information on quality control systems for medicine dispensers operating outside the authorized distribution system (informal sector). These challenges, coupled with the dire lack of studies in these thematic areas [22], warrant future research endeavors to explore the perspectives of cancer patients, as well as the role of the informal sector in the chemotherapy supply chain in Cameroon.

To conclude, our study sheds light on the daily struggles of clinicians in providing high-quality cancer care to their patients and families. Many of these challenges come from (and also impact) the affordability and quality of anticancer medicines, driven largely by weak regulatory capacities and extremely high costs, which amplify the circulation of SF medications in the market. Such challenges as related to drug quality or proliferation of SF anticancer medications impose economic hardships to patients and their families, as well as a significant risk to treatment effectiveness and safety. Together, these factors contribute to adverse outcomes among cancer patients in Cameroon. As a result, understanding the interconnectedness may be crucial for improving cancer treatment outcomes among cancer patients. Addressing these challenges,

however, would require multifaceted approaches implemented in a sequential manner. In the short term, simple strategies like developing practical SOPs and training frontline healthcare workers, including pharmacists, may enhance the detection and management of SF anticancer medications. Similarly, deploying low-cost tools such as the chemoPAD to enhance the detection of SF products at the bedside may prevent patients from receiving a product of dubious quality. The long-term solution lies in strengthening Cameroon's regulatory system by equipping it with the necessary resources and tools to execute its regulatory functions independently. Efforts would also be needed to integrate cancer care and treatment into the country's universal healthcare agenda, with a focus on subsidizing medication costs to enhance treatment accessibility and affordability of medicines, in order not to impair patient safety and treatment quality. Finally, global solidarity and partnerships would be needed to sign access deals to improve the availability and affordability of anticancer care and treatment globally as well as mechanisms to mitigate the circulation of SF drugs in global supply chains.

## Supporting information

**S1 Text. Interview guide 1.**
(DOCX)

**S2 Text. Interview guide_2.**
(DOCX)

**S3 Text. Inclusivity in global research questionnaire.**
(DOCX)

## Acknowledgments

The authors would like to acknowledge all the participants/informants of this study for their selfless contribution. We recognize the contribution of all data collectors for this study, for their key role. We equally thank all the reviewers for this manuscript for tirelessly improving its structure and quality. We would like to thank all the authorities of the surveyed hospitals and pharmacies for allowing us the opportunity to speak with their personnel.

## Author contributions

**Conceptualization:** Yauba Saidu, Andreas Frambo, Armstrong Njuh Nang.

**Data curation:** Ndom Paul.

**Formal analysis:** Masong Christine Makia.

**Funding acquisition:** Marya Lieberman.

**Investigation:** Ndom Paul.

**Methodology:** Yauba Saidu, Andreas Frambo.

**Project administration:** Yauba Saidu, Armstrong Njuh Nang, Ndom Paul.

**Resources:** Yauba Saidu, Andreas Frambo, Marya Lieberman.

**Supervision:** Yauba Saidu, Andreas Frambo, Armstrong Njuh Nang, Ndom Paul.

**Validation:** Marya Lieberman.

**Writing – original draft:** Masong Christine Makia.

**Writing – review & editing:** Yauba Saidu, Masong Christine Makia, Andreas Frambo, Armstrong Njuh Nang, Atalay Mulu, Ayenew Ashenef, Muluneh Benyam, Ibrahim Chikowe, Sonak D Pastakia, Colleen R Higgins, Ndom Paul, Marya Lieberman.

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
