## [Decision Letter · Decision Letter 0]

24 Mar 2025

PGPH-D-24-02954

Highlighting the need for quality anticancer medicines in Cameroon. A qualitative assessment on the perspectives of regulators and first line healthcare providers on the problem.

Dear Dr. Makia,

Thank you for submitting your manuscript to PLOS Global Public Health. After careful consideration, we feel that it has merit but does not fully meet PLOS Global Public Health’s publication criteria as it currently stands. Therefore, we invite you to submit a revised version of the manuscript that addresses the points raised during the review process.

We look forward to receiving your revised manuscript.

Kind regards,

Olivier J Wouters, PhD

Academic Editor

Journal Requirements:

Additional Editor Comments (if provided):

Reviewers' comments:

Reviewer's Responses to Questions

**Comments to the Author**

1. Does this manuscript meet PLOS Global Public Health’s publication criteria?

Reviewer #1: Partly

Reviewer #2: Yes

2. Has the statistical analysis been performed appropriately and rigorously?

Reviewer #1: N/A

Reviewer #2: N/A

3. Have the authors made all data underlying the findings in their manuscript fully available (please refer to the Data Availability Statement at the start of the manuscript PDF file)?

Reviewer #1: Yes

Reviewer #2: No

4. Is the manuscript presented in an intelligible fashion and written in standard English?

Reviewer #1: Yes

Reviewer #2: Yes

Reviewer #1: The authors have presented a well-balanced discussion of an important aspect of access to cancer care in Cameroon, which is still too often overlooked.

Although the methodology seems generally sound and the results mostly support the authors conclusions, some critical information seems to be missing from the manuscript to fully allow me to assess this. Specifically, I would need to see the interview guide used, as well as further details regarding the recruitment strategy and saturation. The lack of included regulators is particularly concerning therein. Lastly, critical information is missing regarding the use of 'key documents'. Kindly see detailed comments attached.

Reviewer #2: Very good introduction. Among the reasons for poorer outcomes in LMICs (lines 65-69), I would cite delayed access to medicines (recent Health Affairs paper by Jouni Kuha).

In the introduction I would present further information about the factors that may impact the quality of pharmaceutical care in a country like Cameroon (ranging from regulatory issues to adequate fiscal space etc.). This would provide a useful background information / framing for interpreting the results section and the myriad factors associated with the quality of care. It was otherwise a bit difficult to follow the results section, where the authors jumped from one issue (like administrative procedures and governance structures) to another (like procurement processes). An overarching framework for how this all fits together would be useful.

The methods would benefit from further description / details. The authors should include the semi-structured interview guide so that readers can better understand the methods and interpret the findings. How was the guide developed? Was it piloted? The lack of adequate detail made it very difficult for the reader to follow the Results and understand the nature of the conversations between the researchers and participants.

Why was there such an uneven number of participants from the various groups (very few regulatory officials)? It’s unclear why the authors describe the results related to quality / affordability at the top of the Results; those results seem like they would fit better in the respective sections.

The description of the governance structure is not really a result – this could be described in the Introduction. Any participant comments about the strengths / weaknesses of this structure / these organizations should remain in the Results (the participants’ comments about these structures). The purely descriptive components / Figure can go in the Intro.

Discussion -- the authors should not refer to their own study as “seminal” – this is usually something judged by readers. I think the authors are trying to make the point that this is the first study to explore these issues in Cameroon, which is something useful to state.

**Do you want your identity to be public for this peer review?** For information about this choice, including consent withdrawal, please see our Privacy Policy

Reviewer #1: **Yes:** Dr. Iris Joosse

Reviewer #2: No

---

## [Decision Letter · Decision Letter 1]

22 Aug 2025

PGPH-D-24-02954R1

Highlighting the need for quality anticancer medicines in Cameroon. A qualitative assessment on the perspectives of regulators and first line healthcare providers on the problem.

Dear Dr. Makia,

Thank you for submitting your manuscript to PLOS Global Public Health. After careful consideration, we feel that it has merit but does not fully meet PLOS Global Public Health’s publication criteria as it currently stands. Therefore, we invite you to submit a revised version of the manuscript that addresses the points raised during the review process.

We look forward to receiving your revised manuscript.

Kind regards,

Olivier J Wouters, PhD

Academic Editor

Journal Requirements:

Additional Editor Comments (if provided):

The authors have done a good job revising their manuscript. In addition to responding to the reviewer’s comments, please also address the following issues.

Please change the title to “Barriers to accessing high-quality cancer medicines in Cameroon: A qualitative study of the views and practices of regulators and front-line health care providers”

Please use the term “front-line” health care providers throughout the paper.

I just want to confirm that all the references in lines 75-83 relate to Cameroon specifically. Why did the authors bring up an example from Ethiopia on lines 84-85? I would consider removing.

Avoid vague statements like “(which is more often than not)” (line 97). Delete, or replace with exact % to make clear the scale of the problem (I think these might be the percentages shown in the next sentence).

Please avoid the use of causal language (for example, “these weaknesses … encouraged the proliferation of SF medicines” in the abstract). Also avoid referring to “our setting”; instead, state “Cameroon” or refer to “the setting” (or “this setting” if it’s already been described)

Avoid the use of bullets in the methods / results. Turn these parts into paragraphs.

Line 157 – an average would be a single number, so do you mean “lasted between 35 and 60 minutes”?

Line 197 - at the start of the Results, repeat the total number of interviewees (57)

Lines 201-207 (and rest of Results). Provide sample size when giving % (over 70% refers to what number out of 57? Or what subset if you’re just referring to the fraction of pharmacists?). Also note that the sentence would read better if you wrote “Over 70% (X/Y) of pharmacists did not consider packaging irregularities as a quality concern.”).

Lines 475-480 – did study 38-40 show causal effects? Avoid overstating the impact of these approaches unless there is conclusive evidence. Instead refer to these CHAI initiatives as examples of efforts to improve access to cancer drugs in Cameroon (not as the “primary drivers” of improvements, which is a strong claim). Avoid language like “remarkable results of improved availability and affordability … and patient outcomes” – instead summarize objective findings from studies and let readers interpret whether these results are “remarkable” or not.

Please also review the paper carefully for grammar, spelling, and overall clarity. The following are some examples; this list is not exhaustive. I would carefully copy-edit the entire manuscript.

Line 34 - “in order” is misspelled as “inorder”

Line 35 – delete the word “some” (same thing in line 126 of the methods)

Line 38 – delete the word “emanating”

Line 39 – delete “which enabled the identification of key themes.”

Line 63 - “10million” is missing a space

Line 65 – change “has been attributed to” to “may reflect”

Line 70 - “showing approximately 30% only of patients…” should read “that only approximately 30% of patients …”

Line 125 – change “as well, a documentary review of some …” to “We also reviewed key administrative reports to provide key background information, triangulate data, and substantiate claims made by interviewees.” (if this is what you meant).

Line 125 – “personnel within the national cancer treatment scene” is somewhat awkward phrasing (especially the “scene” part). Perhaps substitute with something like “stakeholders involved in national cancer treatment efforts”

Line 127-128, the sentence has an incorrect verb construction / comma splice: “Saturation was arrived at once, information received was similar, amongst similar role participants”. Also, this same point is made on lines 173-174. I would not repeat points. Please only describe saturation in one place.

Line 134-136 – this sentence is not grammatically correct

Line 445 – split into two sentences (“supply chain; over 70% of those in private pharmacies interviewed in this study failed to acknowledge …” – or something like that).

Line 448 – “It’s” should be “Its”

Line 462 – delete “In this same line”

Line 462-465 – this is a run-on sentence. Rewrite.

Lines 499-500 – delete “This will have been … to the study” (the sentence just before that is very long – break in two).

Again, this is not an exhaustive list at all. There were many sentences that I found difficult to follow. I would suggest revisiting the full text with a view to improving flow and readability.

Reviewers' comments:

Reviewer's Responses to Questions

**Comments to the Author**

Reviewer #1: All comments have been addressed

publication criteria?

Reviewer #1: Yes

3. Has the statistical analysis been performed appropriately and rigorously?

Reviewer #1: N/A

4. Have the authors made all data underlying the findings in their manuscript fully available (please refer to the Data Availability Statement at the start of the manuscript PDF file)?

Reviewer #1: Yes

5. Is the manuscript presented in an intelligible fashion and written in standard English?

Reviewer #1: Yes

Reviewer #1: Thank you for the opportunity to re-review this manuscript. The authors have integrated most of my previous comments to my satisfaction and provided further explanations that have improved the manuscript, yet I have some further remarks for their consideration (see attachment).

**Do you want your identity to be public for this peer review?** For information about this choice, including consent withdrawal, please see our Privacy Policy

Reviewer #1: No

---

## [Editor Report · Decision Letter 2]

27 Nov 2025

PGPH-D-24-02954R2

Barriers to accessing high-quality cancer medicines in Cameroon. A qualitative study of the views and practices of regulators and frontline healthcare providers.

Dear Dr. Makia,

Thank you for submitting your manuscript to PLOS Global Public Health. After careful consideration, we feel that it has merit but does not fully meet PLOS Global Public Health’s publication criteria as it currently stands. Therefore, we invite you to submit a revised version of the manuscript that addresses the points raised during the review process.

We look forward to receiving your revised manuscript.

Kind regards,

Olivier J Wouters, PhD

Academic Editor

Journal Requirements:

Additional Editor Comments (if provided):

The authors have done a good job revising the manuscript. Could they please address the following minor issues.

There’s a typo in affiliation #2 (Institute). The affiliations don’t follow a consistent format - #5 lists the state (NC), whereas #9 lists the city (Notre Dame). Also please list departments before university names (#3-6 and #8-9). Please check all author names and affiliations again carefully for typos.

Line 69 - change “… disproportionate access to quality medicines [11], delayed access to medicines [12], and high circulation of substandard and falsified anticancer medications in LMICS markets” to “… inadequate access to quality-assured medicines [11-13].”

Avoid acronyms, it makes the text very difficult to read. Spell out all acronyms that are not used throughout the manuscript (such as SF, PPE, IDIs, some of the government bodies - instead of putting IDI with every quote, is it possible to use another convention to make clear that these quotes are from respondents? For example, perhaps the authors could assign #s to the respondent, e.g., “Regulator #3”, “Oncologist #7”, etc.)

Lines 136-147 -- please avoid lists in the main text. Either convert to a narrative format or list the stakeholders in a Box. For example, “The respondents fell into three main groups: (1) regulators and quality-control officials; (2) procurement personnel; and (3) health-care personnel. The regulatory group comprised X, Y, and Z. The procurement group consisted of …” Something along those lines.

Avoid bullet points in the Results (lines 250-256) - please convert into paragraph format. Also please note that you’re missing the end of the sentence on lines 255-256 (“… which in turn result in)
---

## [Editor Report · Decision Letter 3]

8 Jan 2026

Barriers to accessing high-quality cancer medicines in Cameroon. A qualitative study of the views and practices of regulators and frontline healthcare providers.

PGPH-D-24-02954R3

Dear Dr Makia,

We are pleased to inform you that your manuscript 'Barriers to accessing high-quality cancer medicines in Cameroon. A qualitative study of the views and practices of regulators and frontline healthcare providers.' has been provisionally accepted for publication in PLOS Global Public Health.

Best regards,

Olivier J Wouters, PhD

Academic Editor